# Suggestions for Chinese University Freshmen Based on Adaptability Analysis and Sustainable Development Education

**Huaruo Chen [1,*], Ling Ling [2], Yonghui Ma [3], Ya Wen [1], Xiyuan Gao [4] and Xueying Gu [1,*]**

1   School of Education Science, Nanjing Normal University, Nanjing 210046, China; ywen1133@126.com
2   School of Applied Science and Technology, Beijing Union University, Beijing 100020, China;
    wtlingling@buu.edu.cn
3   School of Chemical Science and Technology, Yunnan University, Kunming 650091, China;
    mayonghui@mail.ynu.edu.cn
4   School of Vocational and Continuing Education, Yunnan University, Kunming 650091, China;
    y1054765868@163.com
*   Correspondence: 190601021@stu.njnu.edu.cn (H.C.); 02099@njnu.edu.cn (X.G.);
    Tel.: +86-13599933872 (H.C.); +86-13913017319 (X.G.)

**Abstract:** Based on the theory of sustainable development education, this paper tested the adaptability of contemporary university freshmen. Using the compilation and revision of China's College Student Adjustment Scale (CCSAS). This paper examines five dimensions: personal emotional adaptability, learning adaptability, interpersonal adaptability, university identity, and living adaptability. Using a sample of 640 freshmen (from more than 30 universities) and considering the three dimensions of gender, major, and origin, the paper reaches the following conclusions. Generally, the adaptability of freshmen was positive and their adaptation is good; from the perspective of gender, males and females are different in physiology and psychology; from the perspective of major, there are no significant differences; from the perspective of students' origin, there was no significant difference in general, but there were significant differences in personal emotions and university identity. Based on the above results, this paper emphasizes that to achieve the goal of sustainable development education, universities should begin to pay attention to the adaptability of students as soon as they enter the university. In addition to paying attention to the quality of teaching and learning, universities should also pay attention to the individual differences among students and the related factors that hinder their sustainable development. To ensure that students can meet the challenges of society in the future, the university should cultivate students' awareness of sustainable development and their ability to participate in and develop sustainable practices.

**Keywords:** freshmen; adaptability; sustainable development education

## 1. Introduction

Students' adaptability within sustainable development education is critical for student career success [1,2]. It is inevitable that college freshmen who have just graduated from traditional education to professional education will change many of their habits [3]. During this process, they are sometimes highly adaptive, but most of the time, they will experience repeated failures and setbacks due to maladjustment [4]. In the process of promoting education for sustainable development, if students can adapt to their study and life faster, then education for sustainable development will get twice the results with half the effort [5]. Although schools today are emphasizing education about sustainable development [1,6,7], there is no consensus regarding the concept of sustainable development education,

and a relevant systematic and theoretical approach is lacking [1]. Although it is emphasized that universities should pay attention to sustainable development education as students arrive, this concept is introduced, but then fails to converge with the actual needs and difficulties of college freshmen [8]. Therefore, this paper attempts to understand, first, the adaptability of freshmen and, second, the needs of students in all aspects of university life in order to put sustainable development education into practice and construct individual approaches to sustainable development from the perspective of the students themselves.

### 1.1. Adaptability of Freshmen

When university freshmen transition from high school to university, they experience a phase of life during which they are highly adaptable [9]. From the perspective of whole life development, adaptability during a transition period, as a special form of individual development, embodies the characteristics of specific transition situations and requirements and is also a stage that should be focused on in sustainable development education [10]. Adaptability refers to gradually achieving a dynamic balance with many aspects of a new environment [11]. As the degree of change in the relationship between individuals and the environment in different transition periods differs from the specific requirements of the new environment, the connotation of adaptability in the transition period depends on the specific content of the change in environmental requirements [12]. Therefore, the adaptability of the transition period is situation-specific. In the transition period of university freshmen, many aspects of the individual and social environments have changed such as social role, environment, and separation from original social relations. All of these changes lead to the individual's adaptability during this transition period in aspects such as learning, daily life, interpersonal communication, and so on. In essence, the adaptability of freshmen when they enter university is essentially the psychological manifestation of the interaction between the individual's reconstruction and a specific environment [13]. Therefore, the adaptability of freshmen refers to what occurs after they enter the new environment (from the beginning to the end of the first semester) as they work to achieve a harmonious and balanced state within the requirements of the new environment after they leave their familiar environment [14]. On one hand, freshmen adjust their physical and mental state to meet the requirements of the new environment; on the other hand, they try to change the surrounding conditions of the new environment so that it better suits their psychological characteristics, which is one of the purposes of sustainable development education. Thus, it is essential to propose suggestions for sustainable development education that take advantage of the adaptability of freshmen.

### 1.2. Sustainable Development Education

Education plays an important role in human development. The general assembly of the United Nations announced the International Implementation Plan of the United Nations Decade of Education for Sustainable Development (DESD) for the period 2005 to 2014 and called on governments around the world to integrate sustainable development education into their national education strategies and action plans at all relevant levels during that decade [15]. The ultimate goal of sustainable development education is to enable educated people to effectively solve the environmental and development problems encountered in their daily learning, lives, and work [16]. The educated can actively participate in sustainable development actions, thus achieving the goal of sustainable development education.

At the same time, the DESD notes the priority tasks of the plan and actions of education for sustainable development, which mainly include poverty eradication, gender equality, health promotion, environmental conservation and protection, rural reform, human rights, culture and communication technology, etc. [17]. Based on this task, this paper classifies and analyses the differences in adaptability brought about by these different factors from the perspectives of the gender, major, and origin of freshmen to put forward more accurate suggestions for the implementation of sustainable development education.

## 2. Methodology

The purpose of this paper is to study the adaptability of university freshmen and to propose suggestions to the university based on the theory of sustainable development education.

### 2.1. Procedure

Sustainable development education is a long-term and continuous process [1]. Nurturing the adaptability of university freshmen is the beginning of this process, after which follow-up education can be implemented. This paper proceeds according to the following steps:

- Select the appropriate adaptability scale; modify and determine the content of the scale after the preliminary test.
- To limit the sample, only freshmen can complete the questionnaire.
- SPSS and Amos were used to analyze the sample data, and the adaptability of freshmen was determined.
- Discuss the relationship between sustainable development education and adaptability and emphasize the importance of understanding adaptability to implement sustainable development education.
- Discuss, draw conclusions, and make suggestions.

### 2.2. Subsection

#### 2.2.1. Scale

Baker, R. and Xiaoyi Fang's [1,5] Student Adaptation to College Questionnaire (SACQ) has 60 questions. If the correlation between the scores of each item and the total score of the scale is too low ($r < 0.30$), it means that the item cannot reflect the content of the scale well and should be deleted [7]. After the preliminary stage of the test, there were 24 questions that were not enough to reflect the content of the scale well in this paper, so these 24 questions were deleted in the formal sample survey, 36 questions were revised and determined including eight items on personal emotional adaptability (Table A1), nine items on learning adaptability, seven items on interpersonal adaptability, nine items on university identity, and three items on lifestyle adaptability. See Table 1 for the specific composition of the questionnaire.

**Table 1.** University Students' adaptation scale description.

| Subscale Name | Number of Questions | Subject Number |
|---|---|---|
| 1. Personal Emotional Adaptability | 8 | 1 *,3 *,4 *,7 *,17 *,19 *,25 *,30 * |
| 2. Learning Adaptability | 9 | 2,5,8,14,16,20,23,28,34 * |
| 3. Interpersonal Adaptability | 7 | 6,11,15 *,18,24,31 *,35 * |
| 4. University Identity | 9 | 9,10 *,12,13 *,21 *,22 *,27,32,36 * |
| 5. Living Adaptability | 3 | 26,29,33 * |

Note: *: Reverse question.

#### 2.2.2. Sample

As a sample, this paper randomly selected 48 undergraduate universities including Nanjing Normal University, Nanjing University, Yunnan University, Beijing Union University, Guangxi Normal University, Communication University of China, Inner Mongolia Agricultural University, and Dalian University of Technology as well as 78 majors of first-year freshmen. At the time of sampling, the arts and sciences and genders were balanced as much as possible. A total of 670 test questionnaires were distributed, 665 questionnaires were returned, and the recovery rate was 99.25%. Twenty-five invalid questionnaires were excluded, accounting for 3.7% of the total number of questionnaires. The 640 valid questionnaires are summarized in Table 2.

**Table 2.** Basic data of the sample.

|  |  | Number of People | Percentage |
|---|---|---|---|
| Gender | Male | 292 | 45.63% |
|  | Female | 348 | 54.38% |
| Major | Science | 332 | 55% |
|  | Arts | 288 | 45% |
| Student origin | City | 306 | 47.81% |
|  | Rural | 334 | 52.19% |

The subjects investigated in this paper were limited to four-year undergraduate students excluding specialist students, adult students, and other types of university freshmen; the selected universities covered six basic types of higher education institutions: liberal arts, science, engineering, teaching, agriculture, forestry, and medicine. The schools had a wide geographical distribution, and the types of professional majors taught include Chinese, English, tourism management, hotel management, vocational education, education, computer science and technology, electronic information engineering, software engineering, communication engineering, applied physics, statistics, automation, mechanical design and manufacturing, medicinal chemistry, biotechnology, ecology, Chinese medicine, nursing, art design, and other majors including literature, science, engineering, medicine, agriculture, education, management, art, and other disciplines.

## 3. Results

*3.1. Data Analysis*

### 3.1.1. Exploratory Factor Analysis

Exploratory factor analysis was used to further investigate the structural rationality of the factors [2]. The principal component analysis method was used with the 36 items of the SACQ to analyze the results, which showed that there were nine initial factors with a characteristic root greater than one, but the gravel diagram showed that it was appropriate to extract five factors, which was consistent with our initial scale and revision, and then perform skew rotation analysis. The following is a gravel diagram of the exploratory factor analysis results for the 640 data, as shown in Figure 1.

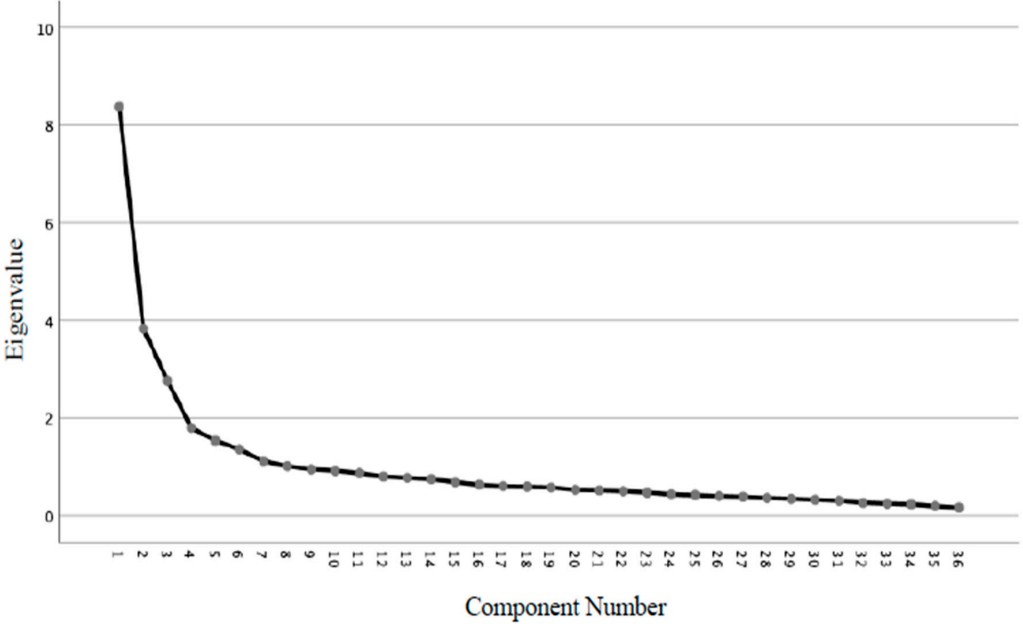

**Figure 1.** Gravel diagram of the exploratory factor analysis of the SACQ scale project.

### 3.1.2. Reliability Level Analysis

Test reliability is an estimate of the degree of consistency of the test results, indicating the stability and reliability of the test. In this paper, it can be used to understand the reliability of the scale by examining the homogeneity reliability (Cronbach's $\alpha$). Reliability was calculated for the entire sample using the Cronbach formula as the internal consistency coefficient of the test. The internal consistency coefficients of each subscale and full scale are shown in Table 3.

**Table 3.** Reliability test results of the official scale (N = 640).

| Subscale Name | Number of Items | Cronbach $\alpha$ |
|---|---|---|
| Total Amount | 36 | 0.869 |
| Personal Emotional Adaptability | 8 | 0.887 |
| Learning Adaptability | 9 | 0.790 |
| Interpersonal Adaptability | 7 | 0.711 |
| University Identity | 9 | 0.848 |
| Living Adaptability | 3 | 0.754 |

As shown in Table 3, the $\alpha$ coefficient of the subscale was in the range of 0.711 to 0.887, and the $\alpha$ coefficient of the total amount was 0.869. Although 0.711 and 0.754 were not very good results, some studies have proven that this value is acceptable and will not lose its explanatory value [7,8,12]. According to the above reliability analysis, the data showed that the scale had high internal homogeneity, and the items of each subscale were conceptually consistent and basically met the requirements of measurement, so the reliability of this scale was better.

### 3.1.3. Validity Analysis

The rationality of the scale structure was analyzed and verified by analyzing the correlation between each subscale and between each subscale and the total scale. According to the theory of psychometrics, each factor of the questionnaire should have a moderate degree of correlation. If the correlation is too high, it means that there are overlaps between the factors, and some factors may not be necessary; if the correlation between the factors is too low, it means that some completely different psychological qualities are measured. The psychologist Tuker pointed out that a good scale structure requires that the correlation between factors and tests is 0.3–0.8, and the correlation between factors is 0.1–0.6. Therefore, the structure validity of the scale is estimated by the correlation between the factors and the correlation between the factors and the total score of the scale (Table 4). The results showed that the correlation between the factors was 0.210–0.528, the correlation between the factors and the total scale was 0.633–0.774, and the correlation was significant ($p < 0.01$). The results showed that the validity of the scale was good and met the requirements of measurement.

**Table 4.** The Correlation between the factors of the scale and the total scale (N = 640).

| Factors | Living Adaptability | Personal Emotional Adaptability | Learning Adaptability | University Identity | Total Scale |
|---|---|---|---|---|---|
| Interpersonal Adaptability | 0.210 ** | 0.417 ** | 0.225 ** | 0.409 ** | 0.631 ** |
| Living Adaptability | | 0.246 ** | 0.456 ** | 0.528 ** | 0.728 ** |
| Personal Emotional Adaptability | | | 0.274 ** | 0.312 ** | 0.671 ** |
| Learning Adaptability | | | | 0.365 ** | 0.633 ** |
| University Identity | | | | | 0.774 ** |

Note: **: $p < 0.01$.

### 3.1.4. Confirmatory Factor Analysis

Five factors obtained from exploratory factor analysis were used as latent variables, and 36 items' scores were used as observation variables to test the correctness of the revised adaptive model of university freshmen, which was completed with Amos 7.0 analysis software (Table 5). The approximate

error index (RMSEA) was less affected by the sample size, and more sensitive to the false model with too few parameters, which is an ideal index, where the smaller the better. RMSEA less than 0.1 indicates a good fit, less than 0.05 indicates a very good fit, and less than 0.01 indicates a very good fit. The final statistical results showed that the approximate error index (RMSEA) was 0.063, indicating that the fitting degree of the model was better. The goodness of fit index $x^2$ can be used with degrees of freedom to show the probability of model correctness. $x^2/df$ is a statistic that directly tests the similarity between sample covariance matrix and estimate variance matrix. The smaller $x^2/df$ is, the better, the closer 1 is, the better the model fitting is, and the model between 2.0–5.0 is acceptable. $x^2/df = 3.153 < 5$, indicating that the model fits well. The relative fit index (NFI, RFI, IFI, TLI, CFI) was more than 0.90, so the model is acceptable. The relative fitting index of the scale was more than 0.90, so it can be considered that the fitting degree of the theoretical model and the data met the statistical requirements, indicating that the theoretical structure of the adaptability of university freshmen was more reasonable.

In conclusion, the revised SACQ has good reliability and validity, and can be used as an effective tool to measure the adaptability of university freshmen.

**Table 5.** Results of confirmatory factor analysis of the scale (N = 640).

| $x^2$ | $df$ | $x^2/df$ | NFI | RFI | IFI | TLI | CFI | RMSEA |
|-------|------|----------|------|------|------|------|------|--------|
| 1841.111 | 584 | 3.153 | 0.963 | 0.958 | 0.975 | 0.971 | 0.975 | 0.063 |

*3.2. Data Results*

This paper used the revised SACQ (to examine the general characteristics of university freshmen's adaptability and differences in demographic variables such as gender, discipline, and student origin) and to compare differences in order to analyze the factors that influence those differences.

3.2.1. General Characteristics of University Freshmen's Adaptability

Table 6 shows the general characteristics of the university freshmen's adaptability, the general description average, and the standard deviation of the overall situation of each factor. The scale project used the Likert score method with a midpoint of 2.5. From the table, it can be seen that the average project had an average score of 2.96, and the average score of each factor also exceeded 2.5, indicating that the overall adaptability of university freshmen and the scores of various factors showed a positive trend. By repeating the measurement of variance analysis of the five dimensions, the in vivo effect test value was extremely significant. According to the results of the post-event comparison test between various factors, the order of the mean values of each factor is as follows: factor five (living adaptability), factor two (learning adaptability), factor four (university identity), factor one (personal emotional adaptability), and factor three (interpersonal adaptability).

**Table 6.** Statistical characteristics of the freshmen's overall characteristics (N = 640).

|  | $M \pm SD$ |
|--|-----------|
| Total Amount | 2.961 ± 1.283 |
| Personal Emotional Adaptability | 2.862 ± 1.293 |
| Learning Adaptability | 3.135 ± 1.110 |
| Interpersonal Adaptability | 2.793 ± 1.273 |
| University Identity | 2.886 ± 1.332 |
| Living Adaptability | 3.325 ± 1.467 |

Note: *M*: Mean; *SD*: Standard Deviation.

### 3.2.2. Gender Differences in University Freshmen's Adaptability

As seen from Table 7, from the gender point of view, on overall adaptability, males scored slightly higher than females, but there was no significant difference. There are, however, significant differences between males and females in terms of personal emotional adaptability and university identity, but at the same time, females' adaptability to learning and interpersonal aspects was slightly lower than that of males, and only the score on living adaptability was slightly higher than that of males. It can be concluded that males and females differ in adaptability in different aspects. Males did better in emotion, while female showed more outstanding results in life ability.

**Table 7.** Gender difference test of university freshmen's adaptability.

|  | Gender | N | $M \pm SD$ | t |
|---|---|---|---|---|
| Total Amount | M | 292 | 2.989 ± 1.309 | 0.234 |
| | F | 348 | 2.939 ± 1.260 | |
| Personal Emotional Adaptability | M | 292 | 2.765 ± 1.302 | 2.466 * |
| | F | 348 | 2.943 ± 1.279 | |
| Learning Adaptability | M | 292 | 3.255 ± 1.149 | 0.873 |
| | F | 348 | 3.035 ± 1.066 | |
| Interpersonal Adaptability | M | 292 | 2.833 ± 1.316 | 0.151 |
| | F | 348 | 2.760 ± 1.234 | |
| University Identity | M | 292 | 2.944 ± 1.326 | −3.020 ** |
| | F | 348 | 2.838 ± 1.335 | |
| Living Adaptability | M | 292 | 3.285 ± 1.560 | 0.476 |
| | F | 348 | 3.358 ± 1.432 | |

Note: M: male, F: female, *: $p < 0.05$, **: $p < 0.01$.

### 3.2.3. Major Differences in University Freshmen's Adaptability

Table 8 shows that from the perspective of the major, there were no significant differences between the arts and sciences in the adaptability of university freshmen including personal emotional adaptability, learning adaptability, interpersonal adaptability, university identity, and living adaptability.

**Table 8.** Major difference test of university freshmen's adaptability.

|  | Major | N | $M \pm SD$ | t |
|---|---|---|---|---|
| Total Amount | Science | 332 | 2.958 ± 1.295 | 0.339 |
| | Arts | 288 | 2.966 ± 1.267 | |
| Personal Emotional Adaptability | Science | 332 | 2.732 ± 1.294 | 1.289 |
| | Arts | 288 | 3.021 ± 1.274 | |
| Learning Adaptability | Science | 332 | 3.191 ± 1.136 | −0.239 |
| | Arts | 288 | 3.066 ± 1.074 | |
| Interpersonal Adaptability | Science | 332 | 2.822 ± 1.306 | −0.274 |
| | Arts | 288 | 2.758 ± 1.229 | |
| University Identity | Science | 332 | 2.914 ± 1.317 | −1.900 |
| | Arts | 288 | 2.852 ± 1.350 | |
| Living Adaptability | Science | 332 | 3.307 ± 1.461 | 1.945 |
| | Arts | 288 | 3.347 ± 1.473 | |

### 3.2.4. Origin Differences in University Freshmen's Adaptability

It can be seen from Table 9 that from the point of view of the origin of students, when considering the university freshmen's adaptive total score and the three factors of learning adaptability, interpersonal adaptability, and lifestyle adaptability, there was no significant difference between freshmen from urban and rural areas. However, in terms of personal emotional adaptability and university identity, there were significant differences in student origin, mainly at the level of personal emotional adaptability,

where university freshmen from cities scored lower than those from rural areas; regarding university identity, the scores of freshmen from the city were higher than that of freshmen from rural areas.

**Table 9.** Origin difference test of university freshmen's adaptability.

|  | Origin | N | $M \pm SD$ | t |
|---|---|---|---|---|
| Total Amount | City | 306 | 2.953 ± 1.304 | 0.866 |
|  | Rural | 334 | 2.969 ± 1.262 |  |
| Personal Emotional Adaptability | City | 306 | 2.768 ± 1.305 | 2.299 * |
|  | Rural | 334 | 2.948 ± 1.276 |  |
| Learning Adaptability | City | 306 | 3.191 ± 1.123 | 1.410 |
|  | Rural | 334 | 3.084 ± 1.096 |  |
| Interpersonal Adaptability | City | 306 | 2.782 ± 1.289 | 0.705 |
|  | Rural | 334 | 2.803 ± 1.257 |  |
| University Identity | City | 306 | 2.911 ± 1.360 | −2.197 * |
|  | Rural | 334 | 2.863 ± 1.305 |  |
| Living Adaptability | City | 306 | 3.257 ± 1.495 | 0.977 |
|  | Rural | 334 | 3.387 ± 1.437 |  |

Note: *: $p < 0.05$.

## 4. Discussion

Based on the theory of sustainable development education, this paper conducted a survey and analysis of the adaptability of university freshmen. The results showed that from the perspective of sustainable development education, university freshmen differed significantly in gender, major, and student origin, and the dimensions of those differences were different. These results also provide a reference for sustainable development education in schools, which should be more comprehensive. Scientifically assessing student differences and intervening appropriately can help achieve the goal of optimal education regarding sustainable development. However, there are still some shortcomings in this paper.

### 4.1. Research Limitations and Deficiencies

Since no research has combined the adaptability test with sustainable development education before, the focus of this paper was to tell schools that only focusing on students' learning was not enough as well as in other areas through the adaptability test results. However, this research was only about a semester of university freshmen's enrolment surveys, and did not track their later adaptability changes. In order to solve this problem, the research will conduct empirical research in the future to test whether it is meaningful for combining the adaptability test and sustainable development education. For this reason, qualitative and quantitative methods are suggested that include, but are not limited to conducting interviews and focus groups, managing surveys, and collecting insights from reflective journals. To assess whether the approach is reasonable, the research will also conduct vertical learning, provide educators with useful information as much as possible, and find the best evidence to help students develop education sustainably.

### 4.2. Long Term Impact of Major Differences

This paper was based on a sample of students representing 78 majors from 48 undergraduate universities. From the above analysis, it was known that there was no significant difference in the adaptability of university freshmen in terms of major, but according to further research in this paper, it was found that in the science sample, 88 people transferred to the arts, accounting for 25% of the population. In the arts sample, only five people transferred to sciences, accounting for 1.7% of the population (see Figure 2 for details). The adaptive problems caused by this part of the population should be significant, but most freshmen in their first semester are studying basic curriculums, and the incompatibility caused by the major differences is not too significant for the time being. Through

in-depth study, especially by increasing their enrolment in professional courses, students will gradually increase their incompatibility. Therefore, because the sample selection in this paper is limited to new students enrolled for only two months, the difference in majors is not significant, and needs to be further studied in subsequent research. This research will also continue to track and report the results in future studies.

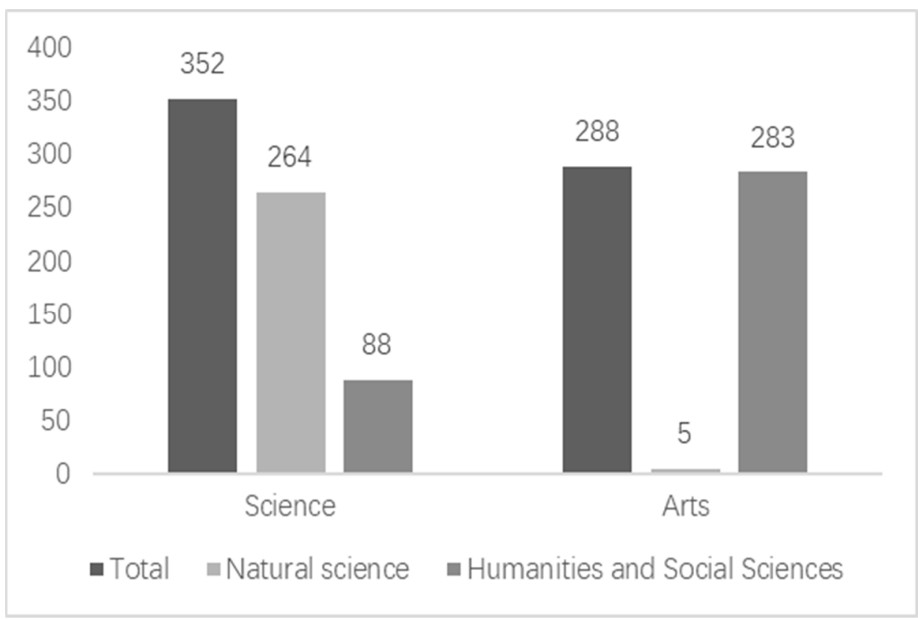

**Figure 2.** Arts and sciences. University professional map.

*4.3. Relevance and Limitations of Adaptability and Sustainable Development Education*

The measurement of adaptability is for people's lifelong development, and sustainable development education is also for people's lifelong development. Based on this purpose, according to the analysis of adaptability results, this paper draws relevant conclusions, so as to question the part of school education that only pays attention to learning. Schools should also pay attention to the education of students in life, personal emotion, and other factors, and all-round education can achieve the purpose of sustainable development education. However, due to the relatively small number of existing studies, the following suggestions in this paper are only based on adaptability to make contributions to sustainable development education, which cannot fully prove that only adaptability can promote the development of sustainable development education, and sustainable development education should be all-round.

## 5. Conclusions

Based on the analysis of the adaptability of university freshmen, the conclusion of this paper is as follows.

*5.1. Overall Differences*

From the above data analysis results, there are some differences in the overall results of university freshmen's adaptability (M = 2.961), gender (M of male = 2.989; M of female = 2.939), major (M of science = 2.958; M of art = 2.966), and origin (M of city = 2.953; M of rural = 2.969). However, even if it can seen from the data results, universities should teach students according to their aptitude and pay attention to the differences between students in different dimensions. However, the current situation of education is that all of the links are separated. Teaching and administrative administration do not cooperate with each other to make effective measures for the students' sustainable development education.

### 5.2. Regional Differences

According to the research in this paper, university freshmen have differences in origin (M of city = 2.953; M of rural = 2.969), which particularly affects their emotions (M of city = 2.768; M of rural = 2.948). It is known that most urban students will have a sense of superiority, and students from rural areas will have insufficient knowledge and a sense of inferiority, leading to the division of social and communication circles after enrolment (M of city = 3.191; M of rural = 3.084). In the implementation process of current education, the concept of "regional equality" is often neglected.

### 5.3. Major Differences

According to the research in this paper, adaptability problems facing freshmen do not differ much, according to the majors they are studying (M of science = 2.958; M of art = 2.966). However, considering the students analyzed in this paper, it is known that with the advancement of university life, the increase in professional courses and professional differences, the significance of those problems will become increasingly obvious (88 people who learned science transferred to the arts, five people who learned arts transferred to sciences).

### 5.4. Gender Differences

According to the research in this paper, there are gender differences in the adaptability of university freshmen (M of male = 2.989; M of female = 2.939), mainly in terms of personal emotions (M of male = 2.765; M of female = 2.943) and university identity (M of male = 2.944; M of female = 2.838). At the same time, there are certain differences in learning (M of male = 3.255; M of female = 3.035) and interpersonal adaptability (M of male = 2.833; M of female = 2.760). After entering university, due to differences in the maturity of mind and body, males and females will have different degrees of differences and problems with both learning and life.

## 6. Suggestions

Based on the analysis and results of the adaptability of university freshmen, this paper combined the definition of the principles of sustainable development education from Tian [1] and constructed an educational model, as shown in Figure 3, in the hope that university education can be combined with the adaptability of university freshmen so that the principles of sustainable development education can be used to optimize student education.

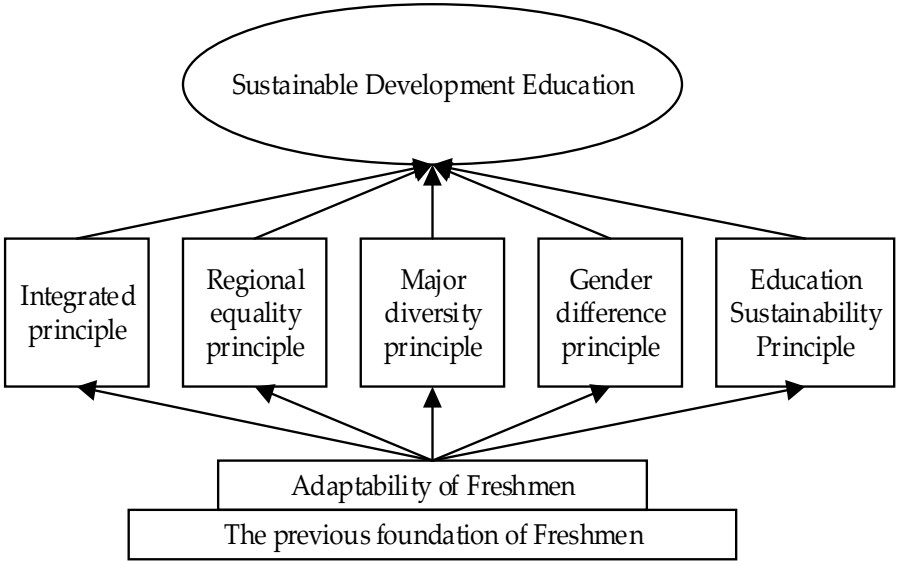

**Figure 3.** Framework of principles for sustainable development education.

### 6.1. Integrated Principle

The integrated principle is the most basic principle in sustainable development education, mainly due to the current positioning of sustainable development education and the characteristics of the individual needs of students. Therefore, adhering to the integrated principle entails systematically considering the factors that influence the sustainable development education of university freshmen (personal emotions, learning, interpersonal relationships, university identity and living); it involves reorganizing, optimizing, and focusing on the influencing factors. It is necessary to maintain the existing overall teaching system and school education model; the review of education and its reconstruction for sustainable development are limited not only to the reduction of course content and adjustment of structural order, but also to the students' own influencing factors [18]. Only by integrating the relationship between what the school provides and what students need can this paper implement sustainable development education in different types of schools and among different types of professional students by paying attention to students' adaptive challenges. Education changes according to the all-round development of the student's mind and body, according to which educational content, educational methods, organizational strategies, and evaluation criteria can be properly adjusted [6].

### 6.2. Regional Equality Principle

It is important to raise awareness of equality between urban and rural areas in sustainable development education. Students' origins often cause them to have difficulties in many aspects of life and can hinder their sustainable development [19]. This type of problem has even caused concern in China about "rural education" and "educational fairness" [10]. Even two people from different origins in the same school will encounter different treatment. If we neglect this principle, it will be difficult to truly advance.

### 6.3. Major Diversity Principle

Different majors are implemented differently in sustainable development education. Everyone will feel the relative success and failure of "sustainable development education". Each of us influences "sustainable development education" through our actions, which may be supportive or destructive. In sustainable development education, the differences in university professional adaptability caused by high school majors require schools to make appropriate adjustments in terms of learning strategies and teaching activities and to pursue appropriate development, according to local conditions [20].

### 6.4. Gender Difference Principle

It is important to focus on gender differences in sustainable development education [17]. After entering university, due to differences in the maturity of mind and body, males and females will have different degrees of differences and problems with both learning and life [21]. Males will have problems such as indulging in games and being unable to take care of themselves. Females are more likely to have problems with dormitory life, relationships, and other issues, and different learning differences will be brought about by the nature of the arts and sciences in academics [22]. Therefore, the school needs to consider the differences raised by gender when paying attention to students' sustainable development education. If necessary, relevant courses should be created, perhaps focusing on relationships and essential life skills [23].

### 6.5. Education about Sustainability Principle

Education about sustainability principles should be integrated into sustainable development education [24]. Sustainable development is a permanent goal as human society continues to develop [25]. It will take several generations—and probably longer—to achieve. Thus, sustainable development education will not be a short-lived educational activity. It is here for the long-term, along with

people's pursuit of sustainable development [26]. From the perspective of education, sustainable development education is regarded as a sustainable development strategy. Therefore, the ultimate goal of sustainable development education is to achieve personal and social sustainable development. From the perspective of educational activities, a school's educational activities must incorporate the idea of sustainable development into every major and promote effective educational methods so that sustainable development education runs through every major, every grade, and everyone's daily life.

**Author Contributions:** H.C. was PI for the project. H.C., Y.M., X.G. (Xueying Gu), and Y.W. developed the questionnaires. L.L. and X.G. (Xiyuan Gao) collected the data. H.C. developed the analytical plan and did the statistical analyses. H.C., L.L., Y.M. and X.G. (Xueying Gu) interpreted the outcomes of the statistical analysis and wrote the paper. All authors have read and agreed to the published version of the manuscript.

**Funding:** This research was funded by Jiangsu Province Basic Education Prospective Teaching Reform Experiment Project and Jiangsu Province University's Advantageous Discipline Construction Project, grant number "PAPD".

**Acknowledgments:** This authors would like to thank Pro. Yao Jijun (working in Nanjing Normal University) for his great support in data analysis and writing revision.

**Conflicts of Interest:** The authors declare no conflicts of interest.

## Appendix A

**Table A1.** The Adaptability Scale of University Freshmen.

| Serial Number | Topic |
| --- | --- |
| 1 | I have been nervous or anxious recently. |
| 2 | In terms of learning, I can keep up. |
| 3 | I have been very depressed recently. |
| 4 | Recently, I have been easily tired. |
| 5 | I am satisfied with my learning situation. |
| 6 | I am satisfied with the extracurricular activities at the university. |
| 7 | Recently, I have not been able to control my emotions very well. |
| 8 | I have a clear learning goal. |
| 9 | Now I am happy because I am at this university. |
| 10 | I think the school atmosphere of this university is very bad. |
| 11 | I get along well with university roommates (if you do not live at the university, please do not answer this question). |
| 12 | I am satisfied with my decision to go to university. |
| 13 | I would prefer to study at another university. |
| 14 | I am satisfied with the number and variety of courses offered at the university. |
| 15 | I feel uncomfortable with my classmates. |
| 16 | I am satisfied with the quality of the courses (depth and breadth) in the university. |
| 17 | Recently, my sleep quality has not been very good. |
| 18 | I am satisfied with the extent of my participation in social activities at the university. |
| 19 | Sometimes my thoughts tend to be a mess. |
| 20 | I am satisfied with the course schedule for this semester. |
| 21 | I prefer to stay at home compared to university. |
| 22 | Recently, I have wanted to go to other schools. |
| 23 | I am satisfied with the way the class teacher teaches. |
| 24 | I am satisfied with my social life at the university. |
| 25 | I encountered many difficulties in dealing with the various pressures at the university. |
| 26 | Parents are not around, I can take care of myself. |
| 27 | I like the campus environment of this university. |
| 28 | I really like my profession. |
| 29 | I am used to the dormitory life of the university (if you do not stay at the university, please do not answer this question). |
| 30 | Recently, I have often had a headache. |
| 31 | I feel that people around me are hard to get along with. |
| 32 | I am satisfied with the university's learning, entertainment, leisure or exercise. |
| 33 | I dare not go shopping alone. |
| 34 | I am not interested in the majors I am studying now. |
| 35 | I am afraid to interact of the opposite sex and the same sex. |
| 36 | I think the school has a negative atmosphere. |

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
