# Peer review of "Suggestions for Chinese University Freshmen Based on Adaptability Analysis and Sustainable Development Education"

_sustainability, doi:10.3390/su12041371_

Round 1

Reviewer 1 Report

In general its a good paper, well structured but Its necessary to improve some methodological aspects:

Line 101: "Sustainable development education is a long-term and continuous process." any reference to to support it. Figure 1 is inteligible, its too small. Table 1: There are some strange characteres ",", whats de meaning of numbers with *? Line 144: "Exploratory factor analysis is used to further investigate the.." any reference to to support it. Line 155 "Cronbach’s a" its necessary use alpha symbol. Conbrachs alpha its not enought to measure reliability. Composite reliability (sometimes called construct reliability) is a measure of internal consistency in scale items, much like Cronbach's alpha (Netemeyer, 2003), and Average Variance Extracted (AVE) its neccesary. Line 160 "data show that the scale has high internal homogeneity", 0.711 its not a very good result, add some reference to support it. Line 172: Authors say that are using Likert scale, It is necessary to establish the graduation of the scale. Table 4: Missing symbol (±) between M SD Table 5: It's necessary to describe information in table adequately. Example of reporting student t "This study found that overweight, physically inactive male participants had statistically significantly lower cholesterol concentrations (5.80 ± 0.38 mmol/L) at the end of an exercise-training programme compared to after a calorie-controlled diet (6.15 ± 0.52 mmol/L), t(38)=2.428, p=0.020."

The discussion is poorly elaborated. When the results are discussed they should be compared with other similar investigations. The section proposed by the authors is more typical of the conclusions. The suggestions section is more typical of the discussion.

Author Response

Point 1: Line 101: "Sustainable development education is a long-term and continuous process." any reference to to support it.

Response 1: Reference added. Thank you for reminding.

Point 2: Figure 1 is inteligible, its too small.

Response 2: Zoomed in on the format in order to make it clearly. In addition, the figure shows that there are five factors in total, which is consistent with the setting of this study from five dimensions. Thank you for reminding.

Point 3: Table 1: There are some strange characteres ",", whats de meaning of numbers with *?

Response 3: I didn’t find characteres "," in Table 1.and added note about meaning of numbers with *. Thank you for reminding.

Point 4: Line 144: "Exploratory factor analysis is used to further investigate the.." any reference to to support it.

Response 4: Reference added. Thank you for reminding.

Point 5: Line 155 "Cronbach’s a" its necessary use alpha symbol. Conbrachs alpha its not enought to measure reliability. Composite reliability (sometimes called construct reliability) is a measure of internal consistency in scale items, much like Cronbach's alpha (Netemeyer, 2003), and Average Variance Extracted (AVE) its neccesary.

Response 5: Thank you very much for your reminding. We have increased the reliability and validity analysis in this paper, and considered the internal fitness of the scale.

Point 6: Line 160 "data show that the scale has high internal homogeneity", 0.711 its not a very good result, add some reference to support it.

Response 6: Reference added, And explains why 0.711 is acceptable. Thank you for reminding.

Point 7: Line 172: Authors say that are using Likert scale, It is necessary to establish the graduation of the scale. Table 4: Missing symbol (±) between M SD Table 5: It's necessary to describe information in table adequately.

Response 7: Table 4: symbol (±) added. Table 5: add more information. Thank you for reminding.

Point 8: Example of reporting student t "This study found that overweight, physically inactive male participants had statistically significantly lower cholesterol concentrations (5.80 ± 0.38 mmol/L) at the end of an exercise-training programme compared to after a calorie-controlled diet (6.15 ± 0.52 mmol/L), t(38)=2.428, p=0.020."

Response 8: It doesn't seem to be the content of this article, we didn’t research exercise-training programme compared to after a calorie-controlled diet.

Point 9: The discussion is poorly elaborated. When the results are discussed they should be compared with other similar investigations. The section proposed by the authors is more typical of the conclusions. The suggestions section is more typical of the discussion.

Response 9: add more information. Thank you for reminding.

All specific changes are displayed in the manuscript. Please check the revised manuscript. Thank you for your comments.

Reviewer 2 Report

The idea presented in this manuscript is interesting. It analyzes how adaptability is utilized into education of sustainable development. However, the connection between adaptability analysis and sustainable development education is weak. It is difficult to find the sustainability aspects within the manuscript. The authors are suggested to submit this manuscript to the other journal if the authors could improve and revise some of the following;

(1)

The title is more attractive if it is revised into “Suggestions for Chinese University Freshmen based on adaptability analysis on sustainable development education”?

(2)

The authors are suggested to include in the Introduction section some references on previous researches of how sustainable development education has been studied so far, what the knowledge/research gap(s) obtained from the previous studies, and what this current research is offering in order to close the gap(s).

(3)

The authors have not presented a strong connection between adaptability and the sustainable education in the manuscript. What the authors have described in the final third of section 1.1 including the whole section 1.2 are mainly talking about adaptability issues instead of sustainable development education. It is more education topics than sustainable ones. The manuscript could be fitted with sustainability issues if the authors provide stronger connection between the adaptability and sustainable development education. The strong connection could be achieved by presenting previous studies on sustainable education issues which need adaptability, if any.

(4)

However, this manuscript is still worth of publication without connection with the sustainability issues if the authors submit it into another journal in education field.

(5)

The authors are suggested to rearrange section 4 and 5 since section 5 contains more explanation than section 5. Section 5 should only contain what the conclusion of the study presented in the manuscript. Therefore, section 5 should be less than section 4. The authors are suggested to read the following material http://jultika.oulu.fi/files/isbn9789514293801.pdf regarding the article elements.

(6)

The authors need to read the guidelines for authors regarding the citation style. Only one style is applicable consistently within a manuscript. Do check and recheck from the guidelines regarding this matter.

(7)

Given the broad readers’ background of this journal, the authors are suggested to provide the digital object identifier (DOI) of each reference in order to facilitate the readers to read the cited references.

Author Response

Point 1: The idea presented in this manuscript is interesting. It analyzes how adaptability is utilized into education of sustainable development. However, the connection between adaptability analysis and sustainable development education is weak. It is difficult to find the sustainability aspects within the manuscript. The authors are suggested to submit this manuscript to the other journal if the authors could improve and revise some of the following;

Response 1: Thank you for your suggestion. This is the first time that we have tried to combine adaptability and sustainable development education. There is no relevant research before. We will further explore the relationship between them in the follow-up research. Hope to get more suggestions from you in the future, Thank you.

Point 2: The title is more attractive if it is revised into “Suggestions for Chinese University Freshmen based on adaptability analysis on sustainable development education”?

Response 2: Thank you very much for your suggestion, which is very good. We also correct the problem in the article.

Point 3: The authors are suggested to include in the Introduction section some references on previous researches of how sustainable development education has been studied so far, what the knowledge/research gap(s) obtained from the previous studies, and what this current research is offering in order to close the gap(s).

Response 3: In the introduction, we summarize the current research, and point out that in the process of implementing sustainable development education in China, we do not pay attention to the adaptability of students. Therefore, this paper attempts to understand, first, the adaptability of freshmen and, second, the needs of students in all aspects of university life in order to put sustainable development opment education into practice and construct individual approaches to sustainable development from the perspective of the students themselves. This is the significance of this article. Thank you very much for your suggestion. I hope my reply can answer your question.

Point 4: The authors have not presented a strong connection between adaptability and the sustainable education in the manuscript. What the authors have described in the final third of section 1.1 including the whole section 1.2 are mainly talking about adaptability issues instead of sustainable development education. It is more education topics than sustainable ones. The manuscript could be fitted with sustainability issues if the authors provide stronger connection between the adaptability and sustainable development education. The strong connection could be achieved by presenting previous studies on sustainable education issues which need adaptability, if any.

Response 4: As there is no research in this field at present, it is difficult for us to gain more mature experience, but we are also aware of this problem, and in the later part, we have closely discuss and make suggestions on the relationship between adaptability and sustainable development education, and we will continue to learn and conduct in-depth research in the follow-up research. Thank you very much for your advice.

Point 5: However, this manuscript is still worth of publication without connection with the sustainability issues if the authors submit it into another journal in education field.

Response 5: Among the 17 themes proposed by OECD, including education and psychology, Sustainability gives us great support and guidance, and we also hope our article can be published in Sustainability. Thank you for your valuable comments.

Point 6: The authors are suggested to rearrange section 4 and 5 since section 5 contains more explanation than section 5. Section 5 should only contain what the conclusion of the study presented in the manuscript. Therefore, section 5 should be less than section 4. The authors are suggested to read the following material http://jultika.oulu.fi/files/isbn9789514293801.pdf regarding the article elements.

Response 6: Thank you very much for your suggestion. We have read the article you provided to us carefully and revised our article at the same time.

Point 7: The authors need to read the guidelines for authors regarding the citation style. Only one style is applicable consistently within a manuscript. Do check and recheck from the guidelines regarding this matter.

Response 7: Thank you very much for your valuable comments. We have read the article carefully again and revised the inconsistency of the citation format in the article.

.

Point 8: Given the broad readers’ background of this journal, the authors are suggested to provide the digital object identifier (DOI) of each reference in order to facilitate the readers to read the cited references.

Response 8: Thank you very much for your suggestion. We have added DOI to the references.

All specific changes are displayed in the manuscript. Please check the revised manuscript. Thank you for your comments.

Reviewer 3 Report

I don't no think is need to present the aim of the research in Method, and also, in Introduction, please do not repeat. What is the reason for each from 60 questions of questionnaire only 36 remain? this 36 is in direction of the 5 areas tracked in the research? please detailed. Also, please present the value of alpha cronbach for the general questionnaire. Please include the code of the ethics commission. For results, please include some correlation between variables, the authors present only descriptive results, it is not enough. Also, the discussion must be extend with other research similar of this study, and please include the limits of this research.

Author Response

Point 1: I don't no think is need to present the aim of the research in Method, and also, in Introduction, please do not repeat.

Response 1: Liv Li (Sustainability’s Editor) proposed to add a discussion on the purpose of adopting this method in the previous manuscript, so we added a description of the purpose in this part. After discussion, we think it is necessary to explain the purpose of connecting the adaptive method with sustainable development education. But we still made some changes according to your opinions. Thank you for your proposal.

Point 2: What is the reason for each from 60 questions of questionnaire only 36 remain? this 36 is in direction of the 5 areas tracked in the research? please detailed.

Response 2: Before our formal investigation, we revised this scale and deleted it according to the data results. The specific reasons have been modified and explained in the paper.

Point 3: Also, please present the value of alpha cronbach for the general questionnaire. Please include the code of the ethics commission.

Response 3: According to the revised article, we have reported the value of alpha cronbach for the general questionnaire and the value of alpha cronbach for sub dimensions in Table 3. And this questionnaire is open, and our final questionnaire is also published in the appendix.

Point 4: For results, please include some correlation between variables, the authors present only descriptive results, it is not enough.

Response 4: We have discussed in the results section that the current research has separated these parts, but the research results show that we need to follow the principle of integration, taking into account the correlation and interaction of several dimensions. Thank you very much for your advice. We have benefited a lot.

Point 5: Also, the discussion must be extend with other research similar of this study, and please include the limits of this research.

Response 5: The discussion have been further described. Thank you for reminding.

All specific changes are displayed in the manuscript. Please check the revised manuscript. Thank you for your comments.

Round 2

Reviewer 1 Report

References do not follow the regulations of the magazine, the last names appear throughout the text.

Line 259-269 Incorrect style

Author Response

Point 1: References do not follow the regulations of the magazine, the last names appear throughout the text.

Response 1: Has been modified, Thank you for reminding.

Point 2: Line 259-269 Incorrect style.

Response 2: Has been modified, Thank you for reminding.

Reviewer 2 Report

The authors have not followed most the comments the reviewer had given. Most of important points, especially in the connection between adaptability analysis and sustainable development education, have not been revised as the central issue in this manuscript. In addition, the in-text citation style has also not been revised according to the guidelines.

Author Response

Point 1: Most of important points, especially in the connection between adaptability analysis and sustainable development education, have not been revised as the central issue in this manuscript.

Response 1: This paper first introduces the background of sustainable development education, and emphasizes the need to pay attention to adaptability in this process. The difference of adaptive measurement results is very necessary for the reform of sustainable development education. In addition, our final discussions and recommendations are closely related to the process of education for sustainable development. We should pay attention to and practice according to the differences reflected by adaptability. However, the third part of this paper is mainly about the results of adaptability, which may weaken the discussion of sustainable development education. However, we have deepened the discussion on adaptability and sustainable development education in the discussion section based on your and the editor's suggestions. If you have better suggestions, please don't hesitate to give us your suggestions. Thank you very much for your advice.

Point 2: In addition, the in-text citation style has also not been revised according to the guidelines.

Response 2: Has been modified, Thank you for reminding.

Reviewer 3 Report

-

Author Response

Point 1: --

Response 1: Thank you for your previous suggestions. Your suggestions have played an important role in improving our paper. Thank you very much.

Round 3

Reviewer 2 Report

The authors have not followed most the comments the reviewer had given.

This manuscript is a resubmission of an earlier submission. The following is a list of the peer review reports and author responses from that submission.